# Cost Associated with Adherence to the EAT-Lancet Score in Brazil

**DOI:** 10.3390/nu17020289

**Published:** 2025-01-15

**Authors:** Thaís Cristina Marquezine Caldeira, Laura Nassif, Taciana Maia de Sousa, Emanuella Gomes Maia, Henrique Bracarense Fagioli, Daniela Silva Canella, Rafael Moreira Claro

**Affiliations:** 1Department of Preventive and Social Medicine, Universidade Federal de Minas Gerais, Belo Horizonte 30130-100, Brazil; 2Nutrition Department, Universidade Federal de Minas Gerais, Belo Horizonte 30130-100, Brazil; lauranassif60@gmail.com (L.N.); rafael.claro@gmail.com (R.M.C.); 3Department of Social Nutrition, Universidade do Estado do Rio de Janeiro, Rio de Janeiro 20550-170, Brazil; tacianamaias@gmail.com; 4Department of Health Sciences, Universidade Estadual de Santa Cruz, Ilhéus 45662-900, Brazil; manugmaia@hotmail.com; 5Faculty of Economic Sciences, Universidade Federal de Minas Gerais, Belo Horizonte 31270-901, Brazil; hbracarense@hotmail.com; 6Department of Applied Nutrition, Universidade do Estado do Rio de Janeiro, Rio de Janeiro 20550-170, Brazil; danicanella@gmail.com

**Keywords:** EAT-Lancet, diet quality, diet costs

## Abstract

Background/Objectives: Food prices are a crucial factor in food choices, especially for more vulnerable populations. To estimate the association between diet cost and quality, as measured by the EAT-Lancet score, across demographic groups in Brazil. Methods: Data from the 2017/18 Household Budget Survey were used to calculate the EAT-Lancet score, comprising 14 components. Scores ranged from 0 (low adherence) to 42 (high adherence), with emphasized components (e.g., vegetables, fruits, legumes) and limited components (e.g., red meat, sugar, eggs). Results were stratified by per capita income, geographic region, and area of residence and compared using linear regression adjusted for high and low costs. In addition, the association between the EAT-Lancet score (and its emphasized and limited components) and diet cost (continuous) was analyzed for the total population and for income tertiles. Results: The mean EAT-Lancet score was 18.65 points (range: 7 to 25) and the mean diet cost was BRL$0.65/100 kcal. Total scores showed no significant difference between low- and high-cost diets. However, limited intake was more pronounced in low-cost diets, while high-cost diets featured emphasized foods such as fruits, vegetables, and seafood. High-cost diets also included sugars and red meat, while unsaturated fats scored higher in low-cost diets. Each one-point increase in the EAT-Lancet score was associated with a BRL$0.38 reduction in cost, driven by lower costs in the Limited component, especially among the lowest-income strata (reductions of BRL$1.58 and BRL$1.55 in the lowest income and middle income tertiles, respectively). However, higher scores for emphasized foods increased costs (BRL$0.89) in the lowest tertile. Conclusions: Higher EAT-Lancet scores were associated with reduced diet costs, likely influenced by the lower Limited component costs in low-income groups. Emphasized foods, however, tended to increase costs, particularly among the lowest-income group. These findings suggest that the role of diet composition plays a significant role in cost differences and underscore the challenges that low-income groups face in accessing affordable, healthy diets.

## 1. Introduction

The EAT-Lancet diet, proposed by the EAT-Lancet Commission, serves as a comprehensive global model for achieving healthy and sustainable dietary patterns. This framework integrates scientific targets for both human health and environmental sustainability, establishing a ’safe operating space’ for food systems that align with the United Nations Sustainable Development Goals (SDGs) and the Paris Agreement. It emphasizes the consumption of vegetables, fruits, whole grains, pulses, nuts, unsaturated oils, while advocating for a reduced intake of red meat, sugar, and ultra processed foods [1]. The EAT-Lancet reference diet aims not only to reduce the burden of noncommunicable diseases but also to addresses major environmental challenges by encouraging shifts in food production and consumption patterns. These shifts are designed to lower greenhouse gas emissions, preserve biodiversity, and optimize land and water use. By integrating health and environmental considerations, this approach underscores the role of dietary transformation in mitigating climate change and ensuring food security for a projected global population of 10 billion by 2050 [1].

However, despite its potential health and environmental benefits, adherence to this diet largely depends on the population’s financial capacity, as food prices are a crucial factor in food choices, particularly for low-income populations [2,3,4]. Globally, economic inequality and the rising cost of food have become significant barriers to achieving healthy dietary patterns. According to data from the Food and Agriculture Organization of the United Nations (FAO), global food prices rose by 11% in 2022, rendering a healthy diet inaccessible to 2.83 billion people [3]. In Brazil, the high cost of the EAT-Lancet diet compared to typical diets, which poses substantial challenges to its adoption, particularly among low-income households [5]. The rising prices of minimally processed and fresh foods exacerbate these challenges, underscoring the intersection of global food insecurity and local socioeconomic disparities [6].

In Brazil, a country marked by pronounced socioeconomic inequality, disparities in income, education, and regional development profoundly shape dietary patterns and access to nutritious foods [7]. Despite being a middle-income country with a diverse agricultural base, structural inequities limit the affordability and availability of healthy food options for large segments of the population. Low-income households, particularly in rural areas and underserved urban communities, face significant financial barriers to adopting healthier diets [7], such as those recommended by the EAT-Lancet Commission. These challenges are compounded by the rising cost of minimally processed and fresh foods [6], increasing reliance on ultra-processed products [8], and the uneven distribution of resources across regions [9]. Understanding the relationship between diet cost and quality provides valuable insights into dietary habits and access to healthy foods, contributing to the development of equitable social policies. This study therefore aimed to estimate the association between diet cost and quality, as measured by the EAT-Lancet score, across demographic groups in Brazil.

## 2. Methods

This is an ecological study that utilized data from the 2017/18 Household Budget Survey (HBS) conducted by the Brazilian Institute of Geography and Statistics (IBGE) [10]. The HBS is a periodic, cross-sectional survey that assesses household consumption, expenditures, and income, providing insight into the living conditions of the Brazilian population.

The HBS sampling followed a complex cluster design with geographic and socioeconomic stratification based on the 2010 Demographic Census and IBGE’s Integrated Household Survey System (SIPD). Sampling proceeded in two stages: first, Primary Sampling Units (UPAs) were selected proportionally to household counts within census sectors; second, households within UPAs were randomly chosen, ensuring seasonal distribution across the survey year. The sample comprised 57,920 households across 575 strata [10]. The use of data by strata allows the use of data with greater geographic and income homogeneity. Furthermore, the interviews were distributed among the strata throughout a year to capture the seasonality of food and its costs.

To ensure the representativeness of the sample and account for population distributions, post-stratification sampling weights were applied. These weights were calculated to adjust for differences in sampling probabilities across strata and were calibrated based on demographic and socioeconomic variables such as age, sex, and household income, derived from population projections aligned with the 2010 Census. The post-stratification weights ensure that the results accurately reflect the population structure and regional variations, mitigating potential biases introduced by the complex sampling design and non-response. This approach is essential to provide robust and generalizable estimates for the Brazilian population. [10]. Further details on the sampling process are available in IBGE publications [10].

### 2.1. Data Collection and Organization

Data were collected from 11 July 2017 to 9 July 2018, and included information on private households and their residents, captured through in-person questionnaires [10]. This study utilized data on food and beverage purchases for home consumption, family income, and household location.

Household food purchases were recorded over seven consecutive days, detailing expenditure, unit price (BRL$/g or BRL$/mL), place of purchase, acquisition type (monetary or non-monetary), and quantity (g or ml). These records were either provided by a household member or collected by an IBGE interviewer. Data were analysed by stratum, allowing for seasonal adjustment, and expressed as daily per capita consumption and proportional expenditure.

Food items were matched to the Brazilian Food Composition Table [11] and categorized by caloric content. Approximately 1800 food and beverage items were grouped into 37 subgroups for analysis, aggregating similar items (e.g., types of bananas). The nonedible fraction of each item, such as peels, shavings, and pits, was removed when appropriate.

Total family income was calculated by combining monthly monetary and non-monetary income of all household members [12]. Monthly per capita income was derived by dividing total household income by the number of household members and categorized into tertiles. Analyses were also stratified by geographic region (North, Northeast, Central-West, Southeast, South) and household area (urban/rural).

### 2.2. EAT-Lancet Score Construction

Adherence to the EAT-Lancet reference diet was assessed using a scoring system developed by Stubbendorff et al. [13]. The EAT-Lancet score included 14 dietary components, split between Emphasized and Limited components. Emphasized components included the food groups of vegetables, fruits, whole grains, legumes, seafood, nuts, and unsaturated oils, while limited components included the food groups of various meats, eggs, dairy products, potatoes, and added sugar (Appendix A). Each component received scores from 0 to 3, with the final score (EAT-Lancet Score) ranging between 0 and 42 points. Details about punctuation are described in Table 1. The higher the final score, the greater the adherence to the EAT-Lancet, and therefore, a healthier diet. Both partial indexes (emphasized and limited components) ranged between 0 and 21 points.

To align with the EAT-Lancet recommendations, the original scoring system used intake quantities expressed in grams for a 2500 kcal/day diet [13] which would be the diet for an adult. For instance, consuming 300 g of vegetables would yield the maximum score of 3 points for that component. This distribution follows the guidelines set by the EAT-Lancet Commission, which defines the recommended intake in grams and the associated caloric contribution for each food group. In this study, we adapted the scoring system by converting the recommended intake in grams to percentages of total caloric intake. This adaptation was necessary due to the absence of directly estimated individual consumption data. We calculated per capita food acquisition by dividing the total household food acquisition reported in the HBS by the number of household residents aged 10 years or older (Table 1). This approach allowed us to estimate individual dietary intake and adjust for varying caloric needs across demographic groups, such as women and children. A similar methodology was employed in a study of the Mexican population in 2024 [14].

### 2.3. Diet Cost Categorization

Diet costs were categorized as “high” or “low” based on the adjusted cost per 2500 kcal. First, we regressed daily diet costs on caloric intake, with residuals reflecting cost variation independent of caloric intake. This method was used to categorize diet costs because of the need to isolate changes in food cost that are not directly attributable to caloric intake. This is important because diets with higher caloric intake naturally tend to have higher costs, even if the quality or nutritional profile remains similar. By regressing daily diet cost on caloric intake, the resulting residuals represent cost variation that cannot be explained by the total calories consumed, allowing for a more accurate analysis of differences in diet costs associated with food composition rather than volume consumed. These residuals were used to create an adjusted cost variable, categorizing diets above the median as “high” cost and those below as “low” cost, removing caloric intake variability.

### 2.4. Statistical Analysis

Proportions and 95% confidence intervals (95% CI) were estimated for the EAT-Lancet score, the Emphasized and Limited components, food groups, and cost per 100 kcal for the total population and stratified by per capita income, geographic region, and household area. Linear regression models were used to evaluate differences in the EAT-Lancet score (and its emphasized and limited components), intake prevalence, and cost per 100 kcal between food groups, adjusting for diet cost, total calories, income (per capita), geographic region, and household area for a 2500 kcal diet. Predicted means with 95% CI were obtained using the “margins” command, and differences between the highest- and lowest-cost diets were tested using a linear combination of parameters. Additionally, the association between the EAT-Lancet score (and its emphasized and limited components) and diet cost was analysed by treating cost as a continuous variable for the total population and for income thirds. These analyses were adjusted for diet cost, total calories, income (per capita), geographic region, and household area. All analyses were performed with significance set at *p* < 0.05. Analyses were performed in Stata version 18.1 (College Station, TX, USA: StataCorp LLC) and weighted for national representation and study design.

## 3. Results

For the total population, the mean EAT-Lancet score was 18.65, with a range of 7 to 25 points. No significant differences were observed in the EAT-Lancet score across income tertiles or by household area. Regional differences were noted, with the South region scoring lower (17.27 points) compared to the North, Southeast, and Northeast regions. The South region also had lower scores for both the Emphasized (4.98 points) and Limited components (10.44 points), while the North region showed the highest score for the Limited components (12.32 points). Among per capita income tertiles, the lowest income tertile scored highest for Limited components (11.69 points), while the highest income tertile had the lowest score (10.94 points) (Table 2).

The average cost of the diet was BRL$0.65/100 kcal for the total population, with variations observed across income levels, geographic regions and household area. A lower cost was seen in the lowest income tertile (BRL$0.60/100 kcal) compared to the highest income tertile (BRL$0.70/100 kcal). Regionally, the Northeast had the lowest cost (BRL$0.58/100 kcal) compared to the Southeast (BRL$0.71/100 kcal), and rural areas had a lower cost (BRL$0.53/100 kcal) than urban areas (BRL$0.69/100 kcal) (Table 2). Visualization maps of the differences between EAT-Lancet scores and diet costs across geographical regions are presented in Figure 1.

When comparing EAT-Lancet scores between low- and high-cost diets, no significant differences were found in the total score (*p* = 0.296) or the Emphasized components (*p* = 0.700). However, lower-cost diets had a significantly higher score for the Limited components (12.03 vs. 10.98; *p* < 0.001). Differences in the score of food groups were observed. Among the emphasized components, vegetables, fruits, and seafood displayed higher prevalence and scores in higher-cost diets, whereas unsaturated fats were more common in lower-cost diets. Among the limited components, beef and lamb, pork, poultry, dairy and potatoes displayed lower prevalence and higher scores in low-cost diets, whereas added sugars were less common in high-cost diets. Whole grains and nuts scored minimally due to low overall acquisition in the population, though nut acquisition was twice as high in high-cost diets (0.17 vs. 0.08, *p* = 0.007). Acquisition of whole grains showed no difference between cost groups (*p* = 0.323). The cost per 100 kcal was generally higher in high-cost diets across most foods in the emphasized intake category, as well as for beef and lamb (Table 3).

For each one-point increase in the total EAT-Lancet score, there was an average decrease of BRL$0.38 in the cost of the diet, while for each one-point increase in middle income tertile of the EAT-Lancet score, there was an average decrease of BRL$1.1 in the cost of the diet. Regarding the Emphasized component, for every one-point increase in lowest income tertile of the score, there was an average increase of BRL$0.89 in diet cost. Finally, for the Limited component, for each one-point increase in the score, there was a reduction in the average of BRL$0.96 for the total score, a reduction of BRL$1.58 for lowest income tertile, and a reduction of BRL$1.55 for middle income tertile (Table 4).

## 4. Discussion

Our study found a mean EAT-Lancet score of 18.65 points. Regional differences were evident, with the South region scoring the lowest for EAT-Lancet score and both the Emphasized and Limited component, while the lowest income tertile scored highest for the Limited component. The average dietary cost was BRL$0.65 per 100 kcal, with lower costs observed among individuals in the lowest income tertile, the Northeast region, and in rural area. Furthermore, each one-point increase in the EAT-Lancet score was associated with a reduction of BRL$0.38 in diet cost. This cost reduction was most pronounced in the Limited component for the lowest income strata, while increased costs in the Emphasized component were observed primarily in the same group.

The regions of Brazil exhibit highly heterogeneous in terms of food culture [15] and also levels of economic development [16], which may help explain the differences in EAT-Lancet scores and diet costs. The cost of diets in rural areas is lower, however, the purchasing power of households in these regions is also reduced. Therefore, food insecurity is significantly more prevalent in rural areas compared to urban ones [3]. This disparity is further highlighted in a multicentre study on dietary intake, which indicates that low fruit and vegetable intake is associated with difficulties in affording these foods [17]. In this context, a basic income could serve as a valuable tool to improve access to food [18], potentially enhancing food security. Regional disparities were also evident, with the southern region presenting lower EAT-Lancet scores compared to other areas. This difference can be partially attributed to dietary patterns characterized by higher consumption of red meat, dairy products [10], and ultra-processed foods [8], which negatively impact the Limited component score. Regarding the Emphasized component, a lower inclusion of cereals, legumes, fish, and nuts likely contributed to the reduced score, despite the region showing relatively higher consumption of fruits and vegetables compared to other regions in Brazil [10]. It is important to note, however, that the southern region is among the wealthiest in the country, with higher average household incomes than other regions [16].

Comparisons between low- and high-cost diets revealed no significant differences in the overall EAT-Lancet score or the Emphasized component. However, the Limited component score was higher in lower-cost diets. High-cost diets tended to include more vegetables, fruits, and seafood, aligning more closely with healthy eating guidelines [1], while unsaturated fats had higher scores in low-cost diets. Lower-cost diets also showed higher scores for foods such as meat, dairy, and potatoes due to lower acquisition rates, whereas added sugars were more frequently consumed in higher-cost diets. Although high-cost diets included greater amounts of fruits, vegetables, fish, whole grains, and nuts, they still contained substantial quantities of added sugars and red meat. This highlights the need for public policies to limit the consumption of these products, even among individuals with greater purchasing power. A study using dietary intake data from Brazil applied a score based on the EAT-Lancet diet (ranging from 0 to 150, with a population mean of 46.4) to evaluate its relationship with food insecurity and income levels. The study found a significant association between severe food insecurity and lower scores, with a 1.31-point reduction compared to individuals with food security. Additionally, low scores were observed for the consumption of nuts, whole grains, and seafood, with scores below 0.5 out of a maximum of 10 points for adequacy. For fruits and vegetables, individuals with higher income and food security demonstrated higher scores [19]. These findings align with our results, which also indicate poor adherence to whole grains and nuts across the population, as reflected in the low EAT-Lancet index scores for these foods.

It is also worth noting that, for low-income populations, the reduced cost of diets is often achieved through the exclusion of expensive and limited foods, such as red meat and dairy products. These results can be attributed to financial constraints that limit access to these foods, leading to a greater reliance on more affordable options, such as staple cereals, beans, and unsaturated oils. This dynamic is particularly evident in the lower cost observed for each additional point in the Limited component between the lowest income tertile (-BRL$1.58) and the middle-income tertile (-BRL$1.55), highlighting how the exclusion of expensive foods, such as meat and dairy, contributes to reducing the overall cost of the diet. On the other hand, the Emphasized components, which include foods such as fruits, vegetables, and fish, showed a high cost, especially in the lowest income tertile, where each additional point in the component was associated with an increase of BRL$0.89. These findings reveal a dual challenge for low-income groups: while lower consumption of restricted foods contributes to lower-cost diets that are aligned with EAT-Lancet recommendations, the higher cost of emphasized foods makes it more difficult to adopt a complete and balanced diet. While this pattern may align more closely with the EAT-Lancet recommendations regarding food restriction, it also highlights potential nutritional vulnerability. Lower-cost diets may lack diversity and show inadequate consumption of emphasized foods, such as fruits and vegetables, which are essential for a nutritionally balanced diet.

These findings align with previous research indicating that the Brazilian population’s dietary quality is low, with high consumption of refined grains and red meat and insufficient intake of fruits, vegetables, nuts, seeds, and whole grains [20]. Although this previous study also employed a different diet quality metric, it similarly found low average dietary quality scores (average score of 14.5 on a scale of 0 to 49 points) [20].

In terms of household food purchases, there has been a marked increase in the acquisition of ultra-processed foods (from 12.6% in 2002/03 to 18.4% in 2017/18) and animal-based foods (from 16.2% to 18.3%) alongside a reduction in the proportion of natural and minimally processed foods (from 53.3% to 49.5%) [10]. Shifts in food prices have likely influenced these changes. Data from the 2008/09 HBS indicate rising food prices, impacting families’ access to certain foods [21]. Recent findings suggest that ultra-processed foods have become more affordable than natural and minimally processed foods in 2022 [6].

Globally, the cost of the EAT-Lancet diet varies significantly based on economic levels. In 2011, the average daily cost of this diet was $2.84, with affordability differences between high-income countries ($2.66) and low-income countries ($2.42). The diet represented 6.1% of daily household income in high-income countries but up to 89.1% in low-income countries [4]. In Brazil, dietary cost also poses a barrier, research shows that adhering to a healthy diet can increase costs by up to 34% for low-income populations when more fruits and vegetables are included, a key component of the EAT-Lancet diet [22].

It is worth noting that when the EAT-Lancet score was analysed against the continuous cost of the diet, a reduction in cost was observed as the score increased. This finding aligns with the results of a Mexican study that employed a methodology similar to ours. In Mexico, a country with an income level comparable to Brazil, each one-point increase in the EAT-Lancet score corresponded to an average reduction of MXN$0.4 in diet cost. The study also reported differences in the EAT-Lancet score between individuals consuming low-cost and high-cost diets (20.3 vs. 19.4 on a scale of 0 to 42; *p* < 0.001), attributing the disparity to lower intakes of beef and lamb, pork, poultry, dairy products, and added sugars among those with lower-cost diets. [14]. Notably, the reduction in diet cost associated with lower consumption of these items was more pronounced among lower-income groups. In contrast, an increase in diet cost driven by higher scores for emphasized food groups—such as fruits, vegetables, seafood, and nuts—was observed primarily within the lowest income tertile. This pattern underscores the dual influence of food group inclusion and exclusion on diet costs: while limited intake of red meat and other restricted items reduces costs, the prevalence of high-cost foods like fruits and nuts tends to drive them up. Similar dynamics have been observed in modelling studies conducted in Brazil using data from the Household Budget Survey (HBS), further corroborating our findings [5,22].

Furthermore, a study by Hirvonen et al. using data from 2011 indicated that the EAT–Lancet diets were more costly in Latin American and Caribbean countries compared to other regions of the world. The largest share of the cost of the diets was for the cost of fruits and vegetables (31.2%), followed by legumes and nuts (18.7%), meat, eggs and fish (15.2%) and dairy products (13.2%). Furthermore, regarding the affordability of these diets, the cost of the EAT–Lancet diet exceeded the total income of at least 1.58 billion people, of whom 80% are in middle-income countries [4]. Access to healthy, sustainable food is further limited by food insecurity, affecting 71.5% of people in low-income countries globally [3]. In 2023, approximately 21.6 million Brazilians faced food insecurity [23]. Income-support programs like Bolsa Família have been instrumental in reducing food insecurity and improving basic food access among Brazilian households [24]. However, improvements in food acquisition patterns vary across regions and subgroups. In highly food-insecure households, not only fruits and vegetable consumption increased, but also cereals, beans, meat, milk, and ultra-processed foods saw substantial increases [25]. This trend highlights the importance of policies that address inequalities in healthy food access and mitigate the effects of price volatility on public health [24,25].

Among the main public actions for healthy eating in the country, the Brazilian Dietary Guidelines represent a milestone in national food policy, aligning with EAT-Lancet recommendations by emphasizing sustainability and promoting the consumption of locally sourced foods, thereby fostering healthier and more sustainable food systems [26].

A promising recent initiative is the revised composition of the basic food basket as outlined in Decree No. 11,936 [27], which aligns with Brazil’s National Policy on Food and Nutrition Security and the National Food Supply Policy, encouraging adherence to the Brazilian Dietary Population [26]. By including healthier food options, such as fresh and minimally processed foods, in the basic food basket, this policy is expected to serve as a benchmark for new regulations aimed at improving the nutritional quality of foods commonly consumed by low-income families.

Furthermore, the use of financial incentives, such as subsidies for natural foods, as well as the imposition of taxes on ultra-processed foods, may represent a pertinent economic intervention, promoting the adoption of healthier and more sustainable diets and mitigating their associated costs [28]. Brazil’s current tax reform [29] represents a significant opportunity to promote affordable and healthier diets. While the reform established zero tax rates for basic food items, it did not extend to ultra-processed foods. However, it made progress by introducing a selective tax on sodas. This measure could serve as a financial disincentive for the consumption of unhealthy beverages, aligning with global recommendations to reduce sugar intake and prevent diet-related chronic diseases.

Together, these policy changes highlight the potential for targeted interventions that not only reduce the cost of healthier food options but also discourage the consumption of ultra-processed products. To maximize their impact, these initiatives should be integrated with public health campaigns that educate the population about the benefits of healthier food choices and address structural barriers, such as limited access to fresh foods in low-income regions.

### Limitations

Despite the significant results, some limitations should be acknowledged. The study relied on data from household food purchases, which may not directly reflect actual individual consumption patterns. This approach assumes a uniform distribution of food among household members and does not account for food consumed outside the home, potentially leading to biases in estimating dietary intake. However, household data collection is an important tool for obtaining information at the population level and can provide robust information for health and nutrition surveillance. Furthermore, the use of a 2500 kcal reference diet allowed adjustments to make the data more accurate and representative. Additionally, the price data used were from 2018, the last available national survey. Since then, Brazil has experienced significant economic changes, including high inflation rates, particularly for fresh and minimally processed foods, and economic disruptions caused by the COVID-19 pandemic. These factors have likely worsened food access for low-income households. The pandemic also disrupted food supply chains, contributing to price volatility and reducing the purchasing power of families, disproportionately affecting access to healthy diets for the most vulnerable populations [6].

Furthermore, it is important to emphasize that the data obtained are based on food acquisition and do not necessarily equate to actual consumption. While domestic food waste, particularly post-preparation, can significantly reduce the quantities of food directly ingested, potentially leading to an overestimation of food availability and dietary costs, correction factors per food [11] were applied in this analysis. These adjustments aim to account for food losses at various stages, including preparation and storage, thereby mitigating, to some extent, the impact of waste on the estimated dietary costs. Nevertheless, it remains important to interpret the results with caution, as waste levels may vary by food group, household size, and socioeconomic status, potentially affecting the relationship between food acquisition patterns and individual dietary intake.

This study is cross-sectional in design, which limits its ability to establish causal relationships between diet costs and dietary quality. While associations were observed, it is not possible to determine whether higher diet costs lead to better adherence to healthy eating patterns or if other factors, such as socioeconomic status, influence both diet cost and quality simultaneously. Future longitudinal studies are needed to explore causal pathways and better understand the dynamics between cost, diet quality, and broader determinants of food choices.

## 5. Conclusions

Our study underscores the intricate relationship between cost and diet quality, highlighting significant demographic disparities. Although the total population did not exhibit direct differences in the overall EAT-Lancet score, foods emphasized for a healthy diet were more prevalent in high-cost diets. Notably, an increase in EAT-Lancet score was associated with a BRL$0.38 reduction in cost, which may be partially explained by lower costs in the Limited component among the lowest income strata. However, higher scores for emphasized foods increased diet costs, particularly in the lowest income tertile, creating a financial barrier to accessing healthier diets.

These findings underscore the influence of diet composition on cost and the challenges of achieving healthy and affordable diets, particularly for low-income groups. Policies that prioritize food subsidies or incentives for healthier foods, such as fruits, vegetables, and legumes, could reduce these barriers. Integrating these findings into national programs like Bolsa Família could enhance the program’s impact on food security and nutrition. For example, expanding conditionalities to include access to affordable healthy foods or directly supporting the purchase of emphasized food groups could help low-income families align their diets with EAT-Lancet recommendations. Such initiatives would promote equity in dietary quality and reduce the economic burden of adopting healthier and more sustainable eating patterns.

## Figures and Tables

**Figure 1 nutrients-17-00289-f001:**
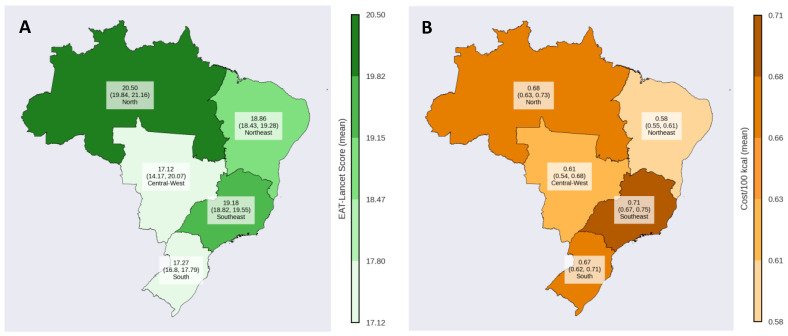
EAT-Lancet score and diet cost across the geographical regions of Brazil. Note: (**A**): EAT-Lancet score (mean and 95% confidence interval) by geographic regions; (**B**): Diet cost/100 kcal (mean and 95% confidence interval) by geographic regions.

**Table 1 nutrients-17-00289-t001:** Criteria for the EAT-Lancet score.

Food Component- EAT-Lancet Diet	Target Intake (and Reference Interval) for 2500 kcal, g/day	kcal/g	Target Intake in %	3 Points	2 Points	1 Points	0 Points
Emphasized components							
Vegetables ^a^	300 (200–600)	0.26	3.12	>3.12	2.08–3.12	1.04–2.08	<1.04
Fruits ^a^	200 (100–300	0.63	5.04	>5.04	2.52–5.04	1.28–2.52	<1.28
Unsaturated oils ^a^	40 (20–80)	8.85	14.16	>14.16	7.08–14.16	3.56–7.08	<3.56
Legumes ^a^	75 (0–150)	3.79	11.36	>11.36	5.68–11.36	2.84–5.68	<2.84
Nuts ^a^	50 (0–100)	5.82	11.64	>11.64	5.84–11.64	2.92–5.84	<2.92
Whole grains ^a^	232	3.50	32.44	>32.44	16.24–32.44	8.12–16.24	<8.12
Seafood ^a^	28 (0–100)	1.43	1.6	>1.6	0.8–1.6	0.4–0.8	<0.4
Limited components							
Beef and lamb ^b^	7 (0–14)	2.14	0.6	<0.6	0.6–1.2	1.2–2.4	>2.4
Pork ^b^	7 (0–14)	2.14	0.6	<0.6	0.6–1.2	1.2–2.4	>2.4
Poultry ^b^	29 (0–58)	2.14	2.48	<2.48	2.48–4.8	4.8–9.6	>9.6
Eggs ^b^	13 (0–25)	1.46	0.76	<0.76	0.76–1.48	1.48–2.92	>2.92
Dairy ^b^	250 (0–500)	0.61	6.12	<6.12	6.12–12.24	12.24–24.48	>24.48
Potatoes ^b^	50 (0–100)	0.78	1.56	<1.56	1.56–3.12	3.12–6.24	>6.24
Added sugars b	31 (0–31)	3.87	4.8	<4.8	4.8–9.6	9.6–19.2	>19.2

Adapted from Stubbendorff et al. [13]. ^a^ direct score; ^b^ indirect score.

**Table 2 nutrients-17-00289-t002:** EAT-Lancet score and diet cost by sociodemographic characteristics.

Groups	Sample			EAT-Lancet Score		Emphasized Intake Score		Limited Intake Score		Cost/100 kcal *	
	*n*	%	95%CI	Mean	95%CI	Mean	95%CI	Mean	95%CI	Mean	95%CI
Total	575	–	–	18.65	18.16–19.15	5.55	5.16–5.92	11.36	11.16–11.56	0.65	0.63–0.68
Income (per capita)											
Lowest income tertile (BRL$739.72)	263	33.42	28.35–38.89	18.78	18.40–19.16	5.61	5.38–5.83	11.69	11.40–11.97	0.60	0.57–0.62
Middle income tertile (BRL$1406.67)	200	33.27	27.93–39.08	18.80	18.40–19.19	5.58	5.54–8.81	11.45	11.14–11.77	0.67	0.63–0.72
Highest income tertile (BRL$2835.32)	112	33.31	26.28–41.17	18.39	17.07–19.71	5.45	4.37–6.54	10.94	10.58–11.30	0.70	0.65–0.75
Geographical region											
North	70	7.02	4.95–9.87	20.50	19.84–21.16	6.46	5.94–6.99	12.32	11.94–12.69	0.68	0.63–0.73
Northeast	191	25.83	21.58–30.58	18.86	18.43–19.28	5.84	5.59–6.08	11.50	11.21–11.78	0.58	0.55–0.61
Southeast	154	39.97	33.75–46.52	19.18	18.82–19.55	5.81	5.54–6.07	11.56	11.23–11.89	0.71	0.67–0.75
South	96	16.01	12.48–20.31	17.27	16.80–17.73	4.98	4.68–5.28	10.44	10.10–10.78	0.67	0.62–0.71
Central-West	64	11.18	5.69–20.78	17.12	14.17–20.07	4.18	1.93–6.43	11.04	10.21–11.87	0.61	0.54–0.68
Household area											
Urban	373	79.53	75.39–83.12	18.79	18.17–19.41	5.62	5.13–6.10	11.30	11.07–11.53	0.69	0.66–0.72
Rural	202	20.47	16.88–24.61	18.13	17.65–18.61	5.28	5.02–5.53	11.57	11.20–11.95	0.53	0.49–0.56

* Values adjusted to 2500 kcal.

**Table 3 nutrients-17-00289-t003:** EAT-Lancet score, prevalence of intake and cost of food groups by diet cost.

	Eat-Lancets Score					Prevalence of Intake					Average Cost per 100 kcal *				
	Low Cost	95%CI	High Cost	95%CI	*p*-Value	Low Cost	95%CI	High Cost	95%CI	*p*-Value	Low Cost	95%CI	High Cost	95%CI	*p*-Value
Total	18.88	18.38–19.38	18.53	18.03–19.07	0.293	-	-	-	-	-	0.63	0.61–0.65	0.67	0.66–0.68	0.007
Emphasized components	5.33	4.98–5.68	5.67	5.30–6.03	0.700	-	-	-	-	-	-	-	-	-	-
Vegetables	0.37	0.28–0.47	0.62	0.56–0.69	0.016	0.92	0.86–0.98	1.11	1.05–1.15	<0.001	1.57	1.50–1.65	1.65	1.58–1.72	0.118
Fruits	1.07	0.91–1.22	1.54	1.41–1.67	<0.001	2.12	1.84–2.40	2.87	2.64–3.11	<0.001	0.75	0.72–0.79	0.81	0.79–0.83	0.007
Whole grains	-	-	-	-	-	0.73	0.60–0.87	0.65	0.55–0.74	0.323	0.33	0.29–0.37	0.34	0.31–0.36	0.853
Legumes	1.22	1.05–1.40	1.02	0.91–1.13	0.087	5.00	4.52–5.48	4.50	4.15–4.85	0.138	0.12	0.11–0.12	0.12	0.12–0.12	0.342
Seafood	0.49	0.37–0.61	0.71	0.61–0.81	0.007	0.43	0.34–0.52	0.58	0.52–0.65	0.011	1.57	1.47–1.68	1.79	1.71–1.89	0.003
Nuts	-	-	-	-	-	0.08	0.044–0.12	0.17	0.16–0.21	0.007	0.66	0.41–0.90	1.07	0.88–1.26	0.005
Unsaturated oils	2.18	2.02–2.34	1.77	1.59–1.95	0.001	12.10	11.19–13.03	9.63	8.76–10.51	<0.001	0.05	0.05–0.06	0.06	0.06–0.07	0.002
Limited components	12.03	11.72–12.35	10.98	10.73–11.23	<0.001	-	-	-	-	-	-	-	-	-	-
Beef and lamb	0.14	0.8–0.21	0.06	0.02–0.10	0.046	4.72	4.18–5.27	5.89	5.21–6.57	0.002	1.06	1.01–1.10	1.12	1.09–1.16	0.038
Pork	1.85	1.67–2.04	1.59	1.42–1.76	0.050	1.11	0.93–1.28	1.40	1.23–1.55	0.023	0.59	0.56–0.63	0.62	0.56–0.64	0.390
Poultry	2.11	1.90–2.28	1.75	1.59–1.91	0.002	3.67	2.97–4.37	4.93	3.95–5.90	0.008	0.61	0.58–0.63	0.61	0.59–0.63	0.951
Eggs	2.39	2.25–2.54	2.35	2.25–2.45	0.647	0.87	0.76–0.98	0.86	0.80–0.93	0.938	0.72	0.69–0.75	0.75	0.73–0.78	0.099
Dairy	2.78	2.71–2.86	2.63	2.56–2.70	0.010	4.78	4.45–5.12	5.63	5.34–5.92	<0.001	0.42	0.40–0.45	0.41	0.39–0.42	0.385
Potatoes	2.74	2.64–2.85	2.59	2.47–2.72	0.044	1.08	0.91–1.25	1.46	1.27–1.65	0.001	0.42	0.40–0.44	0.41	0.40–0.43	0.741
Added sugars	1.52	1.38–1.66	1.88	1.78–1.98	<0.001	10.03	9.25–10.82	7.97	7.46–8.49	<0.001	-	-	-	-	-

Low-cost diet defined as cost residuals below the median (n = 288), and high-cost diet above the media (n = 287), considering a 2500 kcal diet. Median: BRL$6.60; Range for EAT-Lance score goes from 0 to 42 points, with each component going from 0 to 3 points.; * values adjusted to 2500 kcal. All analyses were adjusted for diet cost, total calories, income (per capita), geographic region, and household area.

**Table 4 nutrients-17-00289-t004:** Association of diet cost by EAT-Lancet score and emphasized and limited intake score.

Cost (BRL$)	Coefficient	95% CI	*p*-Value
EAT-Lancet score			
Total	−0.38	−0.75; −0.15	0.042
Lowest income tertile	−0.24	−1.02; 0.54	0.541
Middle income tertile	−1.1	−1.83; −0.38	0.003
Highest income tertile	0.29	−0.05; 0.63	0.090
Emphasized components			
Total	0.25	−0.35; 0.54	0.085
Lowest income tertile	0.89	0.45; 1.32	<0.001
Middle income tertile	0.16	−0.21; 0.53	0.404
Highest income tertile	0.24	−0.48; 0.53	0.120
Limited components			
Total	−0.96	−1.24; −0.67	<0.001
Lowest income tertile	−1.58	2.15; −1.02	<0.001
Middle income tertile	−1.55	−2.09; −1.01	<0.001
Highest income tertile	−0.21	−0.43; 0.01	0.056

All analyses were adjusted for diet cost, total calories, income (per capita), geographic region and household area.

## Data Availability

The dataset supporting the conclusions of this article is available in the IBGE repository [https://www.ibge.gov.br/estatisticas/sociais/saude/24786-pesquisa-de-orcamentos-familiares-2.html?=&t=microdados], accessed on 5 January 2025.

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
