# Peer review of "Cost Associated with Adherence to the EAT-Lancet Score in Brazil"

_nutrients, 2025, doi:10.3390/nu17020289_

Round 1
Reviewer 1 Report
Comments and Suggestions for Authors
This manuscript presents an analysis of the association between diet cost and adherence to the EAT-Lancet score across different demographic groups in Brazil, utilizing data from the 2017/18 Household Budget Survey. The topic is relevant and timely, addressing the important issue of dietary affordability in the context of a sustainable dietary pattern. However, prior to acceptance for publication, the authors should consider the following points:
1. The introduction sets the stage, however, further emphasizing the specific socioeconomic context of Brazil and its relevance to dietary affordability would strengthen the rationale.
2. In the EAT-Lancet Score Construction section, provide more detail regarding the rationale for converting gram data to percentages and any potential limitations of this approach. Mentioning any validation of this methodology would be beneficial.
3. In the Diet Cost Categorization section, elaborate on the justification for using the residual method to categorize diet costs.
4. When discussing the results regarding the drivers of cost reduction, avoid overly definitive language (e.g., "primarily due to"). Consider more nuanced phrasing that acknowledges potential contributing factors.
5. Clarify the statistical significance of the regional differences mentioned.
6. The discussion could benefit from a more in-depth exploration of the reasons behind the observed differences in EAT-Lancet scores and diet costs across income groups and regions.
7. Further discussion on the implications of these findings for public health interventions and policies aimed at promoting affordable healthy diets in Brazil is warranted.
8. Expand on the potential mechanisms driving the association between higher EAT-Lancet scores and reduced diet costs, particularly within lower-income strata.
9. Consider discussing the potential implications of the higher scores for limited components in lower-cost diets, especially concerning nutritional adequacy.
10. In addition to the mentioned limitations, explicitly acknowledge the inherent limitations of using household-level purchase data to infer individual dietary intake.
11. Consider adding the limitation that the study is cross-sectional and cannot establish causality.
12. Mention the potential impact of food waste on the interpretation of cost data.
13. Consider adding a supplemental table providing the specific food items included within the "emphasized" and "limited" components as analyzed in this study.
Author Response
Point-by-point response
Dear reviewers,
We thank you for all your contributions to the advancement of this study. We seek to address all suggestions in the best possible way and to comply with both reviews received.
We remain at your disposal for any clarifications.
Sincerely,
Authors.
Reviewer 1
This manuscript presents an analysis of the association between diet cost and adherence to the EAT-Lancet score across different demographic groups in Brazil, utilizing data from the 2017/18 Household Budget Survey. The topic is relevant and timely, addressing the important issue of dietary affordability in the context of a sustainable dietary pattern. However, prior to acceptance for publication, the authors should consider the following points:
Reviewer 1: 1. The introduction sets the stage, however, further emphasizing the specific socioeconomic context of Brazil and its relevance to dietary affordability would strengthen the rationale.
Authors: We have revised the entire structure of the introduction according to the reviewers' request. We hope to have met your expectations.
Reviewer 1: 2. In the EAT-Lancet Score Construction section, provide more detail regarding the rationale for converting gram data to percentages and any potential limitations of this approach. Mentioning any validation of this methodology would be beneficial.
Authors: To align with the EAT-Lancet recommendations, the original scoring system used in-take quantities expressed in grams for a 2500 kcal/day diet [10]. For instance, consuming 300g of vegetables would yield the maximum score of 3 points for that component. This distribution follows the guidelines set by the EAT-Lancet Commission, which defines the recommended intake in grams and the associated caloric contribution for each food group. In this study, we adapted the scoring system by converting the recommended intake in grams to percentages of total caloric intake. This adaptation was necessary due to the absence of directly estimated individual consumption data. We calculated per capita food acquisition by dividing the total household food acquisition reported in the HBS by the number of household residents aged 10 years or older (Table 1). This approach allowed us to estimate individual dietary intake and adjust for var-ying caloric needs across demographic groups, such as women and children. A similar methodology was employed in a study of the Mexican population in 2024 [11]. We insert this justification in the text.
Reviewer 1: 3. In the Diet Cost Categorization section, elaborate on the justification for using the residual method to categorize diet costs.
Authors: The residual method was applied to categorize diet costs to isolate variations in diet cost not directly attributable to caloric intake. This approach is important because diets with higher caloric intake naturally tend to have higher costs, even when their nutritional profile or quality is similar. By regressing daily diet costs on caloric intake, the resulting residuals represent the variation in cost independent of total calories consumed. This allows for a more precise analysis of differences in diet costs associated with the composition and quality of the diet rather than portion size or caloric quantity. This method ensures that the categorization into "high" and "low" cost reflects qualitative and economic characteristics of the diet rather than simply caloric consumption. We insert this justification in the text.
Reviewer 1: 4. When discussing the results regarding the drivers of cost reduction, avoid overly definitive language (e.g., "primarily due to"). Consider more nuanced phrasing that acknowledges potential contributing factors.
Authors: We have reviewed and updated the terms for greater clarity and more moderate language.
Reviewer 1: 5. Clarify the statistical significance of the regional differences mentioned.
Authors: We have revised the entire discussion structure to adapt to the recommendations made by the reviewers. We hope that we have sufficiently addressed the suggestions.
Reviewer 1: 6. The discussion could benefit from a more in-depth exploration of the reasons behind the observed differences in EAT-Lancet scores and diet costs across income groups and regions.
Authors: We have revised the entire discussion structure to adapt to the recommendations made by the reviewers. We hope that we have sufficiently addressed the suggestions.
Reviewer 1: 7. Further discussion on the implications of these findings for public health interventions and policies aimed at promoting affordable healthy diets in Brazil is warranted.
Authors: We have supplemented the policy implications associated with our findings based on the reviews received.
Reviewer 1: 8. Expand on the potential mechanisms driving the association between higher EAT-Lancet scores and reduced diet costs, particularly within lower-income strata.
Authors: We have revised the entire discussion structure to adapt to the recommendations made by the reviewers. We hope that we have sufficiently addressed the suggestions.
Reviewer 1: 9. Consider discussing the potential implications of the higher scores for limited components in lower-cost diets, especially concerning nutritional adequacy.
Authors: We have revised the entire discussion structure to adapt to the recommendations made by the reviewers. We hope that we have sufficiently addressed the suggestions.
Reviewer 1: 10. In addition to the mentioned limitations, explicitly acknowledge the inherent limitations of using household-level purchase data to infer individual dietary intake.
Authors: We reviewed this section and added the requested limitation. “The study relied on data from household food purchases, which may not directly reflect actual individual consumption patterns. This approach assumes a uniform distribution of food among household members and does not account for food consumed outside the home, potentially leading to biases in estimating dietary intake. However, household data collection is an important tool for obtaining information at the population level and can provide robust information for health and nutrition surveillance.”
Reviewer 1: 11. Consider adding the limitation that the study is cross-sectional and cannot establish causality.
Authors: We reviewed this section and added the requested limitation. “This study is cross-sectional in design, which limits its ability to establish causal relationships between diet costs and dietary quality. While associations were observed, it is not possible to determine whether higher diet costs lead to better adherence to healthy eating patterns or if other factors, such as socioeconomic status, influence both diet cost and quality simultaneously. Future longitudinal studies are needed to explore causal pathways and better understand the dynamics between cost, diet quality, and broader determinants of food choices.”
Reviewer 1: 12. Mention the potential impact of food waste on the interpretation of cost data.
Authors: We reviewed this section and added the requested limitation. “Furthermore, it is important to emphasize that the data obtained are based on food acquisition and do not necessarily equate to actual consumption. Domestic food waste can significantly reduce the quantities of food directly ingested, leading to potential overestimation of food availability and dietary costs. This limitation highlights the need for caution when interpreting cost data, as waste levels may vary by food group, household size, and socioeconomic status, potentially affecting the relationship between food acquisition patterns and individual dietary intake.”
Reviewer 1: 13. Consider adding a supplemental table providing the specific food items included within the "emphasized" and "limited" components as analyzed in this study.
Authors: We have included a table with the foods considered in each category.
Reviewer 2 Report
Comments and Suggestions for Authors
Suggested revisions:
-
Introduction:
- The introduction adequately explains the context, but the role of the EAT-Lancet diet as a framework could be elaborated. Highlight how adherence challenges relate to global patterns of diet cost and inequality (lines 42–59).
-
Methodology:
- Sampling Weights: Clarify how post-stratification weights were applied to account for demographic representation (lines 72–77).
- Score Calculation: Provide detailed justification for adapting the EAT-Lancet score to percentages rather than absolute values (lines 110–112).
-
Results:
- Income Disparities: Elaborate on the significance of higher Limited component scores in low-cost diets, especially among the lowest income tertile (lines 147–153).
- Regional Differences: Provide additional explanation for why the South region scores lower in both Emphasized and Limited components (lines 143–146).
- Statistical Significance: Include more detailed interpretations of the statistical significance of findings for whole grains and other components (lines 168–172).
-
Discussion:
- Global Comparison: Expand the comparison with international contexts, specifically the Mexican study mentioned (lines 259–264).
- Policy Implications: Strengthen the discussion on how findings could inform policies aimed at reducing cost barriers to healthier diets (lines 276–284).
- Limitations: Acknowledge potential biases in extrapolating dietary data from household purchase records to individual consumption patterns (lines 297–300).
-
Figures and Tables:
- Table 2 (lines 157–185): Consider adding a visualization of regional differences and their significance levels.
- Table 4 (lines 182–185): Highlight findings that differ most substantially across tertiles.
-
Language and Formatting:
- Consistency: Use consistent terminology for socioeconomic strata and geographic regions (e.g., lowest-income tertile instead of tertile 1).
- Technical Terms: Simplify or clarify technical terms for broader accessibility (e.g., residuals for diet costs, lines 119–122).
- Spelling and Grammar: Address minor errors, such as “p-valor” instead of “p-value” (lines 182–184).
-
Conclusion:
- Reinforce the policy recommendations by summarizing the most actionable findings. Discuss how these could align with current national programs like Bolsa Família (lines 309–319).
-
References:
- Ensure all references cited are relevant and up-to-date. Cross-check citations, such as reference to [XX] (line 272), and provide the missing source.
Required Changes with Line References
- Line 42–59: Expand the rationale for choosing the EAT-Lancet diet framework.
- Line 110–112: Justify converting dietary recommendations to percentages.
- Line 147–153: Provide insights into higher Limited component scores in lower-income groups.
- Line 143–146: Clarify lower scores in the South region.
- Line 297–300: Address limitations regarding household data and recent economic factors.
Author Response

(The authors gave the same response as above.)

Round 2
Reviewer 2 Report
Comments and Suggestions for Authors
Current version can by accepted. Thank you.